# Cellular and Systemic Effects of Micro- and Nanoplastics in Mammals—What We Know So Far

**DOI:** 10.3390/ma16083123

**Published:** 2023-04-15

**Authors:** Karsten Grote, Fabian Brüstle, Ann-Kathrin Vlacil

**Affiliations:** 1Cardiology and Angiology, Philipps-University Marburg, 35037 Marburg, Germany; 2Stem Cell Unit, Department of Cardiovascular Research, Humanitas Research Hospital, 20089 Milan, Italy

**Keywords:** microplastics, microplastic particles, nanoplastics, nanoplastic particles

## Abstract

Microplastics (MP) and nanoplastics (NP) are accumulating more and more in our environment and have been frequently detected in water and soil, but also in a variety of mainly marine organisms. Polymers such as polyethylene, polypropylene, and polystyrene are those most commonly found. Once in the environment, MP/NP are carriers for many other substances, which often convey toxic effects. Even though intuitively it is thought that ingesting MP/NP cannot be healthy, little is known about their effects on mammalian cells and organisms so far. To better understand the potential hazards of MP/NP on humans and to offer an overview of the already associated pathological effects, we conducted a comprehensive literature review on cellular effects, as well as experimental animal studies on MP/NP in mammals.

## 1. Introduction

Since the beginning of industrial plastic production in the 1950s, the annual production has been steadily increasing, with over 50% of its total 370 billion tons per year now produced in Asia, mainly in China [1]. The basic substances for plastic production are created during the refining process—mostly from crude oil—at low temperatures around 70 °C, and consist of C5–C9 bodies. The resulting plastics are high molecular-weight polymers that can have linear, branched, or cross-linked structures [1]. It is hard to imagine everyday life without products made of plastic because they can be found virtually everywhere and are extremely practical. They can be produced at a relatively low cost, and are easy to shape, stable, and easy to clean if necessary. Their uses range from disposable items such as plastic bags, cups, and packaging material, to synthetic fibers for clothing or paint additives, and even high-quality parts for the electrical and automotive industries. The properties of the end products can vary greatly from very flexible polymers such as polyethylene (PE) or polyester, to very durable polymethyl acrylate (PMA) or polyurethane (PU). The more plastic is produced, the more ends up in the environment, where it is exposed to a wide variety of decomposition processes, for example, erosion due to salt water or mechanical forces. MP are defined as particles in the size range from approximately 1 µm to 5 mm. Larger particles are referred to as mesoplastics and macroplastics, and smaller particles as NP. Studies on the detection of plastics in the environment and their effects on humans and animals have so far largely been carried out for MP, which is also due to the fact that NP are more difficult to detect by metrological means. When MP particles are created directly, they are referred to as primary MP, which are used, for example, in industrial production for cosmetic products, washing and cleaning products, or medical products. Further sources include abrasion from tires and other moving engine parts made of plastic, or when washing clothes made of synthetic fibers. Secondary MP, on the other hand, is formed by the decomposition of macroplastic through environmental processes such as solar UV radiation, wind, waves, temperature, etc. [2]. A good measure of how current or intensely a topic is being worked on is the number of hits in the database PubMed [3] in the U.S. National Institutes of Health’s National Library of Medicine. Using the search terms “microplastics” and “nanoplastics” currently yields ~9200 and ~1550 hits, respectively (as of March 2023). In 2012, there were only a handful of publications on the subject, and this number has risen steadily since then, to ~3400/675 in 2022. The majority of these publications investigate MP physically or chemically, or discuss detection methods or occurrences in soil or water samples. There are also numerous studies on MP in marine organisms, mostly microorganisms of zooplankton. Studies on the detection of MP in these organisms—also in fish—in most cases report negative effects on the biology of these organisms [4]. In recent years, studies on the effects of MP/NP on mammalian cells or on laboratory animals have been increasing. In this review article, we want to focus on such studies in order to be able to better evaluate the possible hazard potential of MP on humans in the future.

## 2. Materials and Methods

This review is based on PubMed searches using the following search terms: microplastics, microplastic (both are abbreviated as MP) particles and microparticles, nanoplastics and nanoplastic (both are abbreviated as NP) particles and nanoparticles. These were combined with the following sub-keywords: intestine/intestinal absorption, inhalation, and skin/dermal penetration. Results were filtered for the hits to the relevant publications in this context.

## 3. Results

MP can take different routes entering a mammalian organism, which include inhalation, ingestion, or uptake via the skin. In the following, we divide the studies according to the different routes by which MP can be absorbed. In this respect, we consider all types of studies, in vitro studies, i.e., cell culture experiments, in vivo studies, i.e., animal studies, and ex vivo studies, i.e., studies on tissues after explantation.

### 3.1. Micro- and Nanoplastic Uptake via the Skin

Given that MP/NP are found in dust, particulate matter, cosmetic products, and as microfibers in many textiles, it is reasonable to consider skin contact as a potential route of MP/NP penetration into an organism. There are particles included in cosmetic products such as body or facial scrubs, but also nanocarriers developed for drug delivery for dermal application that are even intended to abrade or cross the epidermal barrier of our skin. Because of the latter, we already have quite a good understanding of how nanoparticles may cross the biological barrier of the skin. There are two possible methods that can be used, either via the intercellular route following lipid channels between corneocytes or the appendage route via hair follicles or sweat glands.

To be taken up, particles need to penetrate the striatum corneum. Acute and chronic dermatological diseases and injuries including dermatitis, oxidative stress, and inflammation may promote particle uptake. Studies in zebrafish showed impaired skin functions after exposure to polystyrene (PS) MP particles [5], which could possibly facilitate particle uptake.

To study topical administration of nanoparticles to enhance percutaneous drug transport into and across the skin barrier, Alvarez-Román et al. investigated the passive permeation of fluorescently labeled carboxylate-modified polystyrene nanoparticles with a size of 20 and 200 nm across the skin of pigs [6]. The group found that, especially the smaller sized particles, accumulated in the follicular openings of the skin, possibly due to their higher relative surface area compared to the larger particles. However, neither for the 20 nm nor the 200 nm polystyrene nanoparticles was deeper permeation observed and hence, no evidence for uptake via the skin was reported. Similar results were obtained by Campbell and colleagues. By performing in vitro pig skin permeation analysis using laser scanning confocal microscopy, they showed that polystyrene nanoparticles do not penetrate intact or even slightly injured mammalian skin layers, and hardly reach the viable cells of the epidermis, suggesting the skin to be a minor route of exposure to MP/NP [7]. Using two different sizes (70 and 300 nm) of the biodegradable polymer poly(l-lactide-co-glycolide) (PLGA), which is used in medical devices and drug delivery applications, Try et al. found that both particles remained on the skin surface of healthy murine and pig skin, with penetration depths of only 9 and 5–17 µm, respectively [8]. However, using atopic dermatitis models, the inflamed skin of both mouse and pig allowed higher accumulation of 70 nm particles compared to 300 nm sized particles. Of note, in inflamed pig skin the 70 nm sized particles were shown to deeply penetrate into the viable epidermis and/or hair follicles with a penetration depth of about 100 µm. Both murine and porcine atopic dermatitis models revealed a size-dependent penetration depth with 70 nm sized particles showing the biggest effects.

Since animal and human skin show substantial differences, for example, regarding lipid content and hair follicle density [9], data on transdermal penetrability obtained in animal models need to be considered with care. Using ex vivo human skin samples from five different donors, Zou et al. aimed to decipher if and to what extent nanoparticle uptake depends on skin condition, incubation temperature, particle size, and used vehicle solutions [10]. The authors used neutrally charged polystyrene nanoparticles of different sizes (25 nm, 50 nm, and 100 nm), different vehicle solutions (water, dimethyl sulfoxide (DMSO), and ethanol), performed the incubation over 24 h at 4 or 37 °C, and used repeated tape stripping as a simulation for barrier-damaged skin. When using DMSO as a vehicle, the polystyrene nanoparticles penetrated deeper compared to ethanol and water and have been found in the stratum granulosum layer. Incubation at higher temperatures had no effect on skin penetration, and tape stripping also allowed deeper penetration of nanoparticles only until the granulosum layer. Using excised human skin samples, Jatana et al. observed skincare lotion products to improve the access of quantum dot nanoparticles and ingredients such as alpha hydroxyl acids and to facilitate particle penetration [11]. To determine the method of intracellular uptake in a size-dependent manner, Rejman and colleagues used the non-phagocytic B16 murine tumor cell line, which is widely used as a model for human skin cancer [12]. The study shows that fluorescent latex beads smaller than 200 nm were internalized via clathrin-mediated endocytosis, whereas bigger latex beads with a size of 500 nm underwent a caveolae-mediated cellular uptake. Thus, the uptake mechanism and subsequent intracellular routing of MP/NP highly depend on particle size.

Furthermore, for other biological barriers, there is consensus that particle size plays a crucial role. However, given different surface chemistry, polymer origin, and shape, extrapolation of results about skin toxicity and behavior should be performed with care, and further studies on MP/NP are required. In addition to the size, skin condition (intact or compromised) and vehicle solution seem to be among the most important determining properties for the uptake of MP/NP via the skin.

### 3.2. Micro- and Nanoplastic Uptake by Inhalation

Airborne MP/NP can of course not only interact with our skin, but also our lung and pulmonary alveolar epithelium. Exposure may be higher in occupational settings, including for workers in the polyvinylchloride (PVC) flock or synthetic textile industry. Simulating those environments and assessing parameters directly from workers in these industries, adverse respiratory effects have been observed [13,14]. Previously in the 1970s, inhalation of synthetic fibers had been associated with respirational diseases [15,16] and later, lung cancer [17]. Studying aramid, an aromatic polyamide fiber used in the synthetic fiber industry and Kevlar equivalent, Marsh et al. were one of the first to show adverse in vitro effects similar to asbestos [18]. In addition to the occupational exposure in the synthetic fibers and textiles industries, people in cities, especially in large metropolises with much road traffic, are confronted with another source of MP/NP: tire abrasion. According to the Fraunhofer Institute, tire abrasion accounts for 46,000–69,000 tons of MP every year in Germany [19], which consists of a large part of particulate matter.

Tires are made of synthetic rubber, namely polystyrene, which is one of the few types of MP/NP that are commercially available for preclinical toxicological studies. Stimulating human lung epithelial cells with polystyrene MP particles, Dong et al. described the formation of reactive oxygen species (ROS) leading to enhanced inflammatory and cytotoxic effects, as well as a decrease in transepithelial electrical resistance [20]. These observations were confirmed and extended using 40 nm polystyrene NP that decreased human lung epithelial cell viability, induced apoptosis, as well as altered gene expression and altered alveolar epithelial barrier function [21]. Both in vitro studies suggest pulmonary tissue damage followed by potential lung disease after prolonged exposure to polystyrene particles. Using three different sizes of polystyrene microspheres (64, 202, and 535 nm), Brown et al. investigated the effects of particulate matter, especially regarding their size [22]. The group identified the surface area to be a crucial determinant for promoting lung inflammation. Performing bronchoalveolar lavage of rats instilled with microspheres, it was observed that polystyrene particles with a diameter of 64 nm led to increased lavage fluid cell death measured by lactate dehydrogenase activity compared to both control and particles of a bigger size. Performing in vitro experiments using the human adenocarcinoma alveolar basal epithelial cell line A549, the authors reported an increase in Il-8 mRNA expression after 2 h of treatment with the small polystyrene particles. However, this effect was no longer visible when investigating IL-8 protein expression after 2 and 4 h of treatment. Of note, the bigger particles of 202 and 535 nm size showed increased IL-8 protein levels. Plotting the neutrophil response against the particle surface area, the authors identified them to be linked in a proportional manner, illustrating the inflammatory potential of particulate matter. Although this study used artificially high concentrations of polystyrene particles, the revealed surface area mechanism plays a crucial role when considering the ubiquitous presence of particulate matter, especially in urban air and the exposure of susceptible populations.

To address the artificial character of commercially obtained MP/NP (homogenous in size, shape, surface, structure, etc.), Bengalli et al. made use of environmental plastic waste of different origins. After removing impurities and characterization, they applied particles with an average size of 41 µm to stimulate human alveolar epithelial cells. Interestingly, about 72% of the characterized particles were identified as polyethylene. The research group observed induced inflammation, genotoxicity, and cellular uptake of smaller particles [23].

Performing inhalation exposure experiments in mice using tire wear NP particles with a size smaller than 1 µm, Li and colleagues observed diminished ventilatory function and aggravated pulmonary fibrotic injury due to microRNA-mediated inhibition of F-Actin formation [24]. These data suggest that inhaled tire wear MP/NP particles may act as a chronic toxin and provoke pulmonary fibrosis. In addition, another recent in vivo study showed early inflammatory responses in murine lung tissue after inhalation of cationic polystyrene NP. Performing transcriptome analysis, the research group identified the NFκB-NLRP3 inflammasome pathway being activated after exposure, leading to pulmonary toxicity [25]. COVID-19 disposable face masks are mainly made of polypropylene (PP), which can be inhaled after weathering. A recent study investigated the effect of intratracheal installation of 1, 2.5, or 5 mg/kg polypropylene in mice for 4 weeks. Polypropylene NP with an average size of 660 nm induced cytokine and ROS production, as well as increased infiltration of inflammatory cells, suggesting an upregulation of the p38 mitogen-activated protein kinase (MAPK) and NFκB-mediated pro-inflammatory pathways [26]. Assessing the impact of polyethylene terephthalate (PET) NP on human lung carcinoma cells at environmental concentrations, increased oxidative stress, decreased mitochondrial membrane potential, and also cellular internalization can be observed [27].

Taken together, in vitro and in vivo studies using airborne MP/NP particles indicate toxic effects on lung cells and function, including pulmonary inflammation and fibrosis. In the future, susceptible individuals that show impaired removal mechanisms of the lung, e.g., those suffering from chronic obstructive pulmonary disease (COPD), need to be investigated in this regard to identify potentially deteriorating effects caused by the inhalation of MP/NP particles. Again, the particle size and therefore the surface area seems to be the driving factor regarding potentially hazardous effects of inhaled plastic particles.

### 3.3. Micro- and Nanoplastic Uptake by Ingestion

After inhalation, the uptake of MP/NP via food intake and ingestion is supposed to be the second major route of exposure [28]. Of note, MP/NP have been already found in different food and beverages such as drinking water, tea, beer, seafood, or meals served in plastic containers, but also apples and carrots [29]. The presence of MP/NP in edible products is likely, and the number of studies detecting plastic contamination in our food constantly grows because of ubiquitous environmental distribution, air suspension, and precipitation. Hence, it was a groundbreaking but not surprising finding when Schwabl et al. reported for the first time the detection of MP particles in human stool samples in 2019 [30]. As early as 1975, Volkheimer studied the absorption of polyvinylchloride particles and found those MP particles with a size of 5–110 µm in the blood of dogs previously fed with polyvinylchloride powder [31]. However, back then polyvinylchloride was perceived as a very suitable model to study the intestinal uptake of large, solid particles and few people were concerned about the effects their uptake into the organism might have.

To better address the potentially hazardous impact these MP/NP might have when ingested, different studies were performed in vitro or in vivo exposure approaches, using epithelial cell lines or mice that have been administered the plastics via gavage.

Addressing potential gastric effects of ingesting MP/NP, Forte et al. stimulated human gastric adenocarcinoma epithelial cells with polystyrene particles of 40 nm and 100 nm size [32]. Similar to observations made in the skin and lung, also in gastric cells the smaller particles accumulated faster and to a greater extent compared to the bigger ones. The 40 nm particles, especially, provoked inflammatory gene expression and decreased cell viability. These findings are relevant regarding human gastric physiology and toxicology, being among the first tissues to encounter MP/NP food contaminants and also pharmaceuticals, including particle-based drug delivery systems. Xu et al. orally exposed mice to 1 mg polystyrene NP particles with a size of about 100 nm for 4 weeks and observed their accumulation in different organs including spleen, liver, lung, and brain. Additionally, those tissues showed induced cell apoptosis, inflammation, and structure disorders. Using the intestinal epithelial Caco-2 cell line, the research group showed absorption of polystyrene NP particles by macropinocytosis and clathrin-mediated endocytosis accompanied by the disruption of epithelial tight junctions [33].

Co-culturing Caco-2 along with HT-29 cells as a cell culture model for mucus secretion with Raji-B cells, which are lymphocytic cells able to transform Caco-2 into “M” cells characteristic of Payer’s patches, Domenech et al. investigated a physiological/structural in vitro model of the human intestinal barrier [34]. The research group reported a dose-dependent uptake and translocation of polystyrene particles, supporting the observations that MP/NP are able to cross the epithelial barrier of the digestive system. However, no cytotoxic, genotoxic, or altered oxidative effects were observed. In contrast, implementing an inverted cell culture model consisting of Caco-2, HT29, and monocytic THP-1 cells, the research group of Busch et al. simulated the interaction between these cells. In this case there were buoyant polyethylene particles, which more resemble the conditions in the intestine [35]. Interestingly, compared to the conventional, gravity-driven sedimentation in the in vitro model, the research group found cytotoxic and pro-inflammatory effects, including increased interleukin (IL)-1β and IL-8 cytokine release using the inverted model. The authors’ own unpublished data showed that 500 nm polystyrene particles can cross a confluent monolayer of epithelial HEK-293 cells and that a mix of 200, 500, and 1000 nm polystyrene particles increased the permeability of that monolayer for small tracer dyes.

To better address the chronic nature of exposure to MP/NP through our gastrointestinal tract, Domenech et al. also investigated the effects of 50 nm sized polystyrene particles on Caco-2 cells lasting for 8 weeks at sub-toxic concentrations: 0.0006, 0.26, 1.3, and 6.5 µg/cm^2^. Of note, the authors stated that the lowest concentration corresponds to “an estimate of the human ingestion of 7 μg of plastic particles with the consumption of 225 g of mussels” [36]. After 8 weeks of exposure, 20% of the cells showed uptake and internalization of the polystyrene particles and altered expression of oxidate stress-related genes can be detected; however, no genotoxic effects were observed.

A study led by the German Federal Institute for Risk Assessment investigated the effect of polystyrene MP particles with a size of 1, 4, and 10 µm both in vitro and in vivo using Caco-2 epithelial cells, THP-1 monocytic cells, and a mouse feeding study [37]. Besides the observed cellular uptake also seen in previous studies, no adverse effects regarding inflammatory responses or differentiation into and activation of human macrophages were observed. This suggests that exposure to polystyrene particles is not likely to cause adverse health effects.

In contrast, Herrala and colleagues found adverse effects on cell viability, cytotoxicity, and oxidative stress response in both Caco-2 and HT-29 cells stimulated with polyethylene particles at different concentrations (0.25–1.0 mg/mL) and a size range of 5–60 μm [38]. Stimulation with polyethylene particles elevated mitochondrial superoxide production in a concentration-dependent manner. However, no other markers of oxidative stress were found to be altered. To mimic extraction procedures and answer the question whether the particles themselves or leaching chemicals induced cytotoxic effects, the authors also used polyethylene particles extracted with ethanol, which is typically used in toxicity studies of materials that are in contact with food products. However, no effects were observed, demonstrating that the observed cytotoxic effects were caused by the particles themselves.

The partly controversial results mentioned in those studies can be explained by tremendous differences in the experimental setup, ranging from the chosen model to the applied particle concentration and size and exposure route. However, consensus exists regarding the translocation of MP/NP particles from the intestine into the circulation and subsequent cellular uptake, resulting in their accumulation in different organs.

During recent years, the importance of our gut microbiome became more and more evident, being a crucial player in both physiological and pathological conditions [39]. To study the impact of daily exposure to polyethylene MP particles on the microbiome composition and potential adverse effects on metabolites, Fournier et al. applied the in vitro mucosal artificial colon model (M-ARCOL) using stool samples of healthy adult volunteers [40]. Exposure to polyethylene particles resulted in donor-dependent effects and a shift in abundance from beneficial bacteria towards potentially harmful pathobionts. Additionally, no metabolite-mediated impact of polystyrene MP particles on the intestinal barrier was observed. Further studies are needed to assess the extent of translocation and the impact of ingested MP/NP particles in more vulnerable populations including infants (exposed e.g., through plastic teething toys), people on long-term antibiotics treatment, or Crohn’s disease patients.

### 3.4. Micro- and Nanoplastics Can Cross Well-Controlled Barriers of the Body

In this section, we cover important barriers in the body that are particularly tightly regulated to prevent uninvited guests from entering or to remove them again: the blood–brain barrier, the placenta barrier between mother and fetus, and the kidney.

Using 3 different sizes of polystyrene MP particles, 0.2, 2, and 10 µm, Kwon et al. observed accumulation in microglial cells after oral treatment in mice, demonstrating the ability of MP/NP to cross the blood–brain barrier [41]. The same study reported impaired microglial morphology, immune response, and apoptosis provoked by 0.2 and 2 µm particle size in human microglial HMC-3 cells, again revealing the size-dependent effects of MP/NP. Both the human cell line and the mouse brain microglia showed activation of pro-inflammatory cytokines and apoptotic markers, suggesting severe effects of MP/NP particles on cellular processes in the brain. Similar results were observed using polystyrene nanoparticles with a size of 50 nm [42]. Similar to the results were seen regarding skin permeation, where the capability of particles to cross the blood–brain barrier is also size-dependent, with NP showing a higher penetrance compared to MP [43]. However, exposing mice over 180 days to polystyrene particles of different sizes, Jin et al. observed a concentration-, but not size-dependent neurotoxicological phenotype with hippocampal inflammation accompanied by cognitive and memory deficits [44]. Of note, all used particle sizes (0.5, 4, and 10 µm) were detected in the brain of exposed mice.

When polystyrene MP/NP particles with sizes of 100 nm and 1 µm were administered orally to female mice during gestational days 1 to 17, Yang and colleagues observed not only the transfer of plastic particles from the maternal mice to the fetus across the placenta but also serious effects on the fetal thalamus, which in turn induced anxiety-like behavior in the progeny [45]. Combining transcriptomic and immunofluorescence analysis of the fetal brain, the authors suggested that polystyrene nanoparticles induce ROS-mediated oxidative stress, altered γ-aminobutyric acid (GABA)-ergic neurotransmission, and apoptosis, resulting in the inhibition of fetal brain development. Interestingly, the study observed that the larger MP particles (1 µm) promoted the smaller NP (100 nm) to enter the fetal brain and thereby aggravate their neurotoxic effects. Accordingly, by exposing pregnant rats to 20 nm sized polystyrene particles via inhalation at gestational day 19, Fournier et al. observed maternal lung-to-fetal tissue NP transfer also in late-stage pregnancy [46]. Furthermore, exposing pregnant mice to polystyrene NP particles of 100 nm via drinking water was associated with diminished fetal growth as well as impaired cholesterol metabolism in both the placenta and fetus [47]. Performing laser direct infrared spectroscopy analysis, Zhu and colleagues [48] detected in all of the 17 investigated human placentas MP particles at a range of 0.28–9.55 particles/g. Interestingly, with about 80%, the majority of detected MP were smaller than 100 μm in size, pointing again towards a size-dependent uptake. Additionally, about 43% of the detected MP particles were identified as polyvinyl chloride, followed by polypropylene (15%), and polybutylene succinate (PBS, 11%).

Investigating mouse ovarian tissue in vivo, Huang et al. found a decreased number of growing follicles and pups per litter, as well as increased oxidative stress and apoptosis after oral treatment with polystyrene nanoparticles (low dose group: 5 mg/kg/day, high-dose group: 25 mg/kg/day for 8 weeks) [49]. The polystyrene particles with a size of 50 nm were shown to enter the ovarian and uterine tissue, affect sex hormone serum levels, and led to an imbalance of oxidative and antioxidative serum markers, especially at high concentrations. Similar effects were observed stimulating the human ovarian granulosa cell line COV434 with increasing concentrations of the same polystyrene nanoparticles (50, 100, 150, and 200 μg/mL for 24 h), resulting in diminished viability, increased apoptosis, and oxidative stress. The authors identified nuclear factor-E2-related factor 2 (Nrf2) to have a mitigating effect on polystyrene-induced oxidative stress.

Investigating both HEK-293 cells and a murine in vivo model, Li et al. showed that polystyrene NP can synergistically aggravate lipopolysaccharide (LPS)-induced renal cell apoptosis and the expression of the apoptosis markers Caspase-3 and Caspase-12 as well as endoplasmic reticulum stress through oxidative stress [50].

Using a kidney-testis microfluidic platform, Xiao and colleagues addressed the effect of polystyrene nanoparticles on a multi-organ scale in vitro to answer the question if exposure to MP/NP may also trigger cancer-related signaling pathways [51]. Polystyrene nanoparticles were found to enter the kidney and testis via endocytosis and stimulate proteins involved in cancer-related signaling pathways including MAPK and phosphatidylinositol-3-kinase and protein kinase B (PI3K-AKT). The authors claim that polystyrene NP particles exerted potential carcinogenic effects on the human kidney and testis.

### 3.5. Impact of Micro- and Nanoplastics on Immune Cells

In the last section, we turn to the effects of MP/NP on immune cells that are known to date. Independent of the entry route via the skin, lung, or intestine, MP/NP may encounter immune cells such as macrophages, which are resident in almost all organs and tissues, including those discussed here and which are responsible for innate immune mechanisms [52].

As mentioned earlier, food containers are a potential source of exposure of our food to different types of MP/NP, especially when heated up, including polyethylene terephthalate, polyethylene, polypropylene, and polystyrene [53]. This exposure was even exacerbated by the increased use of plastics in food preparation during the COVID-19 pandemic [54].

Focusing on these MP/NP released from food containers, a research group simulated this scenario by collecting leachate from heated commercial plastic products [55]. After characterization, those particles were dyed and further used for in vitro stimulation experiments of mouse RAW264.7 macrophages cells with 4 μg/mL of ~192 nm polyethylene terephthalate, ~1.85 μm polyethylene terephthalate, and low-density polyethylene particles. Performing different imaging techniques, the study shows cellular ingestion, especially for polyethylene terephthalate-derived NP, resulting in a significantly reduced lysosomal activity, playing a crucial role in the generation of ROS. These observations emphasize the importance of finding alternative packaging materials as well as the potential connection of MP/NP-induced disruption of lysosomal activity and the well-known characteristics of lysosomal storage and neurodegenerative disorders such as Huntington’s disease [56,57].

Exposure of human macrophages to sulfate-modified NP with a size of 200 nm resulted in an increased lipid accumulation and foam cell formation accompanied by impaired lysosomal clearance and mitochondrial stress [58]. Of note, foam cell formation is a hallmark for the initiation and progression of atherosclerosis, the underlying pathogenesis of cardiovascular disease [59]. Once in the circulation, MP/NP may interact with blood cells but also vascular cells including the endothelium. Investigating polystyrene nanoparticles as potential drug carriers, Barshtein et al. observed a size- and concentration-dependent increase in red blood cell aggregation and adhesion to endothelial cells [60]. Furthermore, a study by our own research group showed that polystyrene MP particles with a size of 1 µm induced endothelial cytokine and adhesion molecule expression, leading to enhanced adhesion of leukocytes both under static and flow conditions, indicating aggravated endothelial inflammation [61]. In addition, we observed an increased inflammatory response of monocytic cells upon stimulation with polystyrene particles as well as their uptake by neutrophils in the blood. In 2022 Leslie et al. reported for the first time the presence of MP particles in human blood samples, a fact that was not surprising but rather long-awaited due to analytical difficulties [62]. Given the detoxifying nature of our liver, it is reasonable to hypothesize there are also detrimental effects of MP/NP particle exposure on hepatic functions. Investigating primary rat hepatocytes and human hepatocyte cell lines, Johnston and colleagues found that fluorescent polystyrene carboxylated particles with a size of 20 nm were internalized earlier and to a greater extent compared to particles with a size of 200 nm [63]. The size-dependent uptake in hepatocytes is in accordance with in vivo findings [64]. Observing an enhanced cellular uptake of particles when dispersed in a serum-containing medium, the authors suggested either serum protein adsorption onto the particle surface or improved dispersion of the single particles as the underlying cause. Of note, it was shown that protein affinity of polystyrene nanoparticles depends on particle concentration and hydrophilicity, and protein adsorption triggers particle aggregation [65,66]. Thus, the protein corona around different particles is another crucial parameter regarding the availability and uptake of MP/NP.

Investigating 1 µm polystyrene particles with and without the commonly used plasticizer bisphenol A, Cheng et al. reported adverse effects on hepatic metabolism [67]. Liver organoids derived from human pluripotent stem cells indicated increased hepatotoxicity when stimulated with both polystyrene particles and bisphenol A, uncovering synergistic effects. Genes involved in lipid metabolism showed altered expression patterns, making further studies necessary regarding the effect of MP and plasticizers on hepatic lipid metabolism and potential associations with liver steatosis.

Screening different types of MP/NP including polystyrene, polyethylene terephthalate, polyacrylonitrile (PAN), or nylon fibers, only amine-modified polystyrene (PS-NH2) directly activated the NLRP3 inflammasome in THP1-derived macrophages [68]. This innate immune response mechanism plays a crucial role in chronic intestinal and lung inflammatory diseases. However, stimulation with polyethylene terephthalate, polyacrylonitrile, and nylon fibers resulted in the induction of the pro-inflammatory cytokine IL-8, independent of NLRP3, suggesting an alternative route of action.

The mentioned in vivo and in vitro studies suggest detrimental effects of MP/NP on immune and hepatic cells, once those particles have crossed the biological barrier. Since many immune cells circulate in the body, it is conceivable that they release MP/NP particles taken up in the blood or gastrointestinal epithelium in other organs or tissues, e.g., through apoptotic processes. In addition to the damaging effects of the MP/NP particles on the immune cells themselves, damaging effects can then also occur in these tissues due to the discharged particles, e.g., in the liver or spleen. To our knowledge, however, no study has yet been able to show this aspect.

Ranging from neurotoxicity to impaired metabolic pathways during pregnancy or altered inflammatory response and cell cycle signaling pathways that may induce tumorigenesis, more studies are needed to confirm and investigate further the potential hazardous impact of MP/NP particles. The lifelong exposure and accumulation of MP/NP particles might have detrimental, yet unknown, effects that promote and aggravate various chronic diseases and cancer. We have listed all studies showing the polymer used and particle size (Table 1).

### 3.6. General Limitations of Studies with Micro- and Nanoplastics

There are some limitations to studies on MP/NP that should not be ignored when evaluating the recent findings.

First of all, it must be noted that exposure of the cells or animal to MP/NP is of course also present under control conditions and can hardly be avoided. Cell culture vessels, pipette tips, and many other utensils for cell culture work are made of plastic and release MP/NP into the medium. In animal experiments, the situation is similar: the animals are kept in plastic cages, drink from plastic bottles, and the feed is constantly exposed to plastic surfaces in the course of the production and many other processes. Completely MP/NP-free conditions are therefore simply not feasible.

The MP/NP particles that were used experimentally were either purchased from a commercial distributor or produced in-house. In both cases, these particles are suspended after production or aliquoted as a pure substance into appropriate small containers. The particles are not always sterile, but are at least only minimally contaminated with other substances. The situation is quite different with MP/NP from the environment. Many parameters can affect the behavior of MP/NP, for example external environment including temperature, pressure, and pH, but also cell type, size, surface chemistry, or interaction with the lipid mesophase, which all may influence the toxicological potential of these particles, and are nicely described in a review article by Beddoes and colleagues [70]. The particles have a so-called eco-corona due to the adhesion of numerous substances, e.g., nitrogen oxides, proteins, bacterial/viral products, toxic organic chemicals, or heavy metals [71]. It can be assumed that the load of MP/NP with adhering substances depends on the place of their origin and their residence time in the environment. To give an example of how the above-mentioned external environment can affect the biocompatibility of particles, Lundqvist et al. reported that silica nanoparticles in human blood are abundantly surrounded by proteins including serum albumin, immunoglobulin G, or fibrinogen. Making the determination of availability and toxicology of MP/NP even more complex, the protein-corona was different when using whole blood with or without ethylenediaminetetraacetic acid (EDTA), plasma, or serum [72]. For pathogen-associated substances alone, the induction of numerous inflammatory processes via recognition by Toll-like receptors for immune response has been described [73]. Although thorough studies and reviews exist regarding the cellular entry, and there is consent that particles develop an external identity depending on their environment, covered in a hard and soft corona through adsorption of proteins and small molecules including amino acids and sugars, the true importance of the corona is yet to be determined [70]. Additionally, further determining the method of intracellular uptake (phagocytosis, pinocytosis, clathrin- or caveolae-mediated endocytosis) will tell us also about the subsequent intracellular fate. However, a comprehensive assessment of possible risks from these mentioned free riders on MP/NP particles is beyond the scope of our review article. Through the absence of such an eco-corona under laboratory conditions, the observed effects of MP/NP are presumably milder than they would be for MP/NP as they occur in the environment. Additionally, the surface charge can also influence the properties of the particles, e.g., their uptake [69].

One more point needs to be mentioned here. Plasticizers (phthalate) are often added to plastics to prevent the material from being brittle. Even if the plasticizers evaporate over time, after degradation and erosion, these plasticizers can still be found in MP/NP. Just as we have known for a long time that particulate matter—of which MP/NP represent a part—is a risk factor, e.g., for ischemic heart disease, stroke, and chronic obstructive pulmonary disease [74], we already know that plasticizers are a risk factor, e.g., for neurodegeneration, asthma, type 2 diabetes, and cancer [75,76,77,78]. Another potential hazard arising from phthalate esters are abnormal sexual development and birth defects [79,80]. A detailed look at this extensive topic is also not the focus of our review article and is only briefly touched on here.

Experimentally, polystyrene particles are most frequently used, mainly because these particles are commercially available from various producers (e.g., Kisker Biotech (Steinfurt, Germany), Alpha Nanotech Inc. (Vancouver, BC, Canada), CD Bioparticles (Shirley, NY, USA), Spherotech Inc. (Lake Forest, IL, USA)). Although polystyrene is one of the most frequently detected polymers in soil and water samples or marine organisms, polyethylene and polypropylene are often detected at even higher concentrations in these samples. In addition, up to 10 other plastic species can be detected in such samples [81]. At the lower end of the scale of polymers detected in the environment are alkyds and polyurethanes, about whose effects little is known. In this context, it is metrologically much easier to detect MP/NP in soil or water samples using available methods such as Raman spectroscopy, Fourier transform infrared spectroscopy, or pyrolytic methods than in organisms. For example, hemoglobin and myoglobin outshine the spectrum of plastic particles in spectrometric measurement methods. In animal experiments, fluorescence-labeled particles are therefore often used for tracer experiments. In air or soil samples, it is also difficult to make a statement about the concentration of small MP and NP particles in general, as these are only detected to a limited extent by the device’s collector due to their small size and low occurrence, despite long measurement times. How high the exposure of different tissues to MP/NP in humans is, is largely unknown. It is assumed that we ingest up to 5 g of MP per week [82], but this number corresponds to a model calculation of experimental data and not to data from clinical studies. Therefore, the dosage of the particle concentration in the individual experiments is determined empirically and sometimes varies considerably. Mostly, only one or a few particle sizes are investigated in the studies assembled here, and more rarely a mixture of different sizes are used. In real life, however, there is always exposure to particles of different sizes and materials. Regarding NP, the data available are even more limited, which might be due to the even more difficult handling of the particles. Particles in the nm range are often already below the light microscopic detection limit. In most experiments, it also remains open whether the observed effects are MP/NP effects or particle effects, since controls with particles from other materials are usually not performed. Therefore, experimental studies certainly only incompletely reflect reality—which we do not yet know very well—as size, material, etc., of the MP/NP found in our environment are much more heterogeneous than those used experimentally. However, from the existing studies, we have clear indications about which particle properties pose the greatest hazard. The particle size and thus, surface area, seem to play a crucial role regarding skin permeation, uptake by lung, gastric and intestinal epithelial, liver, and immune cells. However, the vehicle in which the MP/NP particles are administered determines their penetration, as seen, for example, in skin permeation experiments where DMSO clearly increased particle uptake [10]. Similarly, the protein adsorption and protein corona around MP/NP particles plays an important role as well, which in turn was shown to be dependent on carboxyl or sulfate modifications [65,66].

## 4. Conclusions

Even if the health risks for the particulate matter and plasticizers associated with MP/NP are already known and the above-mentioned limitations are taken into account, the studies conducted on the subject to date provide a clear picture. Indeed, the vast majority of studies find that MP/NP particles activate inflammatory signaling pathways, induce apoptosis, and have other cytotoxic effects. In addition, it is now quite clear that MP/NP particles can overcome barriers in the body and thus penetrate and accumulate in organs and tissues. For the sake of clarity, we have summarized the most important findings of the studies in a scheme (Figure 1). So far, nothing is known about the late effects of such processes. Together with the findings from marine ecosystems and the increasing pollution of waters, soils, and air, a turnaround is urgently needed. We need better recycling processes including a circular economy for plastics and alternative natural materials to reduce plastic production, and thus MP/NP pollution. There are already some promising examples of the use of natural materials, including car tires made of natural rubber or the use of cellulose fiber-based lyocell for the textile industry. However, none of these products has yet reached the stage where they are ready for series production on an industrial scale. With over 1,000,000 tons in 2019, Germany is the largest exporter of plastic waste in the European Union [83], mostly to low-income countries. Although the amount is constantly decreasing, the lack of an international legal framework makes it easy for high-income countries, which also have the highest consumption of plastic products, to exploit low-income countries as cheap dumping grounds. The uncontrolled burning of plastic waste, especially, is a big problem because large amounts of MP/NP are directly released into the atmosphere and the oceans. Therefore, the export of plastic waste needs to be tightly regulated to prevent the problem of waste disposal from only being shifted among countries. Political responsibility, a critical debate in our society about own consumption and waste avoidance, recycling or the use of alternative materials needs to be enforced. It is therefore the responsibility of the high-income nations to stop the export of waste and to promote the aforementioned processes. Even if it were possible in the foreseeable future to reduce plastic production, improve recycling, and use alternative materials, it would probably take many decades for the burden of MP/NP on the environment, and thus also on animals and humans, to decrease noticeably. Thus, further studies are needed to assess the potential hazardous impact this man-made environmental factor can exert on us and the organisms in our environment.

## Figures and Tables

**Figure 1 materials-16-03123-f001:**
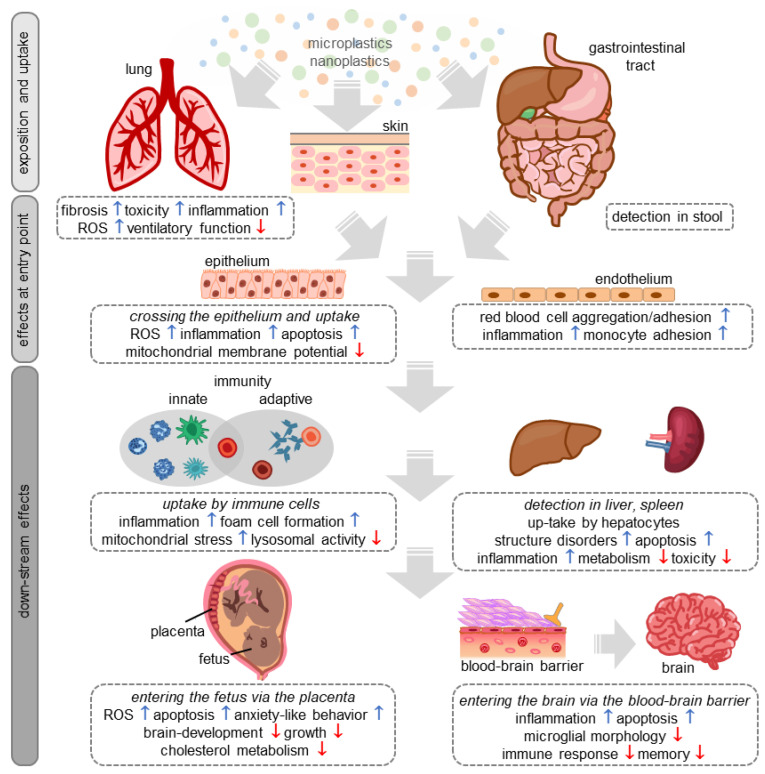
The schematic shows the different uptake pathways and effects of micro- and nanoplastics on different tissues, organs, and cells in mammals. ROS = reactive oxygen species.

**Table 1 materials-16-03123-t001:** Studies investigating the effects of micro- and nanoplastics on mammals (in vitro and in vivo).

Year	Author et al., (reference)	Polymer	Particle Size	Particle Concentration	Manufacturer
2023	Woo [26]	polypropylene	~66 µm	1, 2.5, 5 mg/kg, *A* 1, 2, 4 mg/mL, *B*	in-house
	Fournier [40]	polyethylene	1–10 μm	2.625 mg/mL, *B*	Cospheric LLC (Goleta, CA, USA)
	Wu [25]	polystyrene (amine)	100 nm	5 mg/kg, *A*	Tianjin Saierqun Technology Co. Ltd. (Tianjin, China)
	Cheng [67]	polystyrene	1 µm	10–50,000 ng/mL, *B*	TBCTRC (Tianjin, China)
	Chen [47]	polystyrene	100 nm	1, 10 mg/L, *A*	Shanghai Huge Biotechnology Corporation (Shanghai, China)
2022	Bengalli [23]	waste plastic granules	<50 μm	0.1, 1, 10, 100 μg/mL, *B*	in-house
	Yang [43]	polystyrene	100 nm; 3, 10 µm	200 mg/kg, *A*	Thermo Fisher Scientific (Waltham, MA, USA)
	Jin [44]	polystyrene	0.5, 4, 10 µm	100, 1000 µg/L, *A*	TBCTRC (Tianjin, China)
	Busch [68]	polystyrene (amine)	50 nm	0–50 μg/cm^2^, *A*	Sigma Aldrich (Schnelldorf, Germany), Polysciences Inc. (Warrington, PA, USA)
		polyvinyl chloride	235 nm	0–50 μg/cm^2^, *A*	Werth-Metall (Erfurt, Germany)
		polyethylene	611 nm	0–50 μg/cm^2^, *A*	Cospheric LLC (Goleta, CA, USA)
		polyethylene (terephthalate)	16 nm; 5.7 µm	0–50 μg/cm^2^, *A*	in-house
		polyester	17.5 × 10 µm	0–50 μg/cm^2^, *A*	in-house
		polyacrylonitrile	18.5 × 10 µm	0–50 μg/cm^2^, *A*	in-house
		polyamide (nylon)	27.5 × 10 µm	0–50 μg/cm^2^, *A*	in-house
	Deng [55]	polyethylene (terephthalate), polypropylene, polystyrene	~192 nm, 1.85 µm	4 µg/mL, *B*	in-house
	Yang [45]	polystyrene	100 nm, 1 µm	10 mg/mL (1 mg/d), *A*	TBCTRC (Tianjin, China)
	Shan [42]	polystyrene	50 nm	0.5–50 mg/kg, *A*	Bangs Laboratories, Inc. (Fishers, IN, USA)
	Florance [58]	polystyrene (sulfate)	0.2 µm	100 µg/mL, *B*	Polysciences Inc. (Warrington, PA, USA)
	Li [24]	tire wear microplastic particles	<40 μm	0.125, 0.5, 1 mg/kg, *A* 25, 50, 100 µg/mL, *B*	in-house
	Zhang [27]	polyethylene (terephthalate)	122–221, 142–296 nm	30, 300 μg/mL, *B* 1967.9 μg/mL in different dilutions, *B*	Zhongxin Plastics Co. Ltd. (Shanghai, China)
	Kwon [41]	polystyrene	0.2, 2, 10 µm	2.5, 10 µg/mL, *A* 1, 5, 10 µg/mL, *B*	Spherotech Inc. (Lake Forest, IL, USA)
2021	Yang [21]	polystyrene	40 nm	8–128 μg/mL, *B*	Shanghai Huge Biotechnology Corporation (Shanghai, China)
	Busch [35]	polyethylene	200–9900 nm	0–50 μg/cm^2^, *B*	Cospheric LLC (Goleta, CA, USA)
	Vlacil [61]	polystyrene	1 µm	0.54, 54 ng/mL, *B* 5.4 µg/mL, *B*; 2.5 mg, *A*	Kisker Biotech GmbH (Steinfurt, Germany)
	Domenech [36]	polystyrene	50 nm	6.5, 13, 26, 39 μg/cm^2^ short term, *B* 0.0006, 0.26, 1.3, 6.5 µg/cm^2^ long term, *B*	Spherotech Inc. (Lake Forest, IL, USA)
	Xu [33]	polystyrene	100 nm	10 mg/mL, *A* 30–480 µg/mL, *B*	TBCTRC (Tianjin, China)
2020	Fournier [46]	polystyrene	20 nm	8.8 × 10^14^, *A, C*	NanoCS (New York, NY, USA)
	Domenech [34]	polystyrene	0.04–0.1 µm	1–200 µg/mL, *B*	Spherotech Inc. (Lake Forest, IL, USA)
	Dong [20]	polystyrene	1.7–2.2 μm	1–1000 µg/cm^2^, *B*	in-house
2019	Stock [37]	polystyrene	1, 4, 10 μm	1 × 10^4^–1 × 10^9^, *B* 1.25, 25, 34 mg/kg, *A*	Thermo Fisher Scientific (Waltham, MA, USA), Kisker Biotech GmbH (Steinfurt, Germany)
2017	Zou [10]	polystyrene	25, 50, 100 nm	no further details, *C*	Thermo Fisher Scientific (Waltham, MA, USA)
2016	Try [8]	poly (L-lactide-co-glycolide)	70, 300 nm	no further details, *A*	in-house
2015	Forte [32]	polystyrene	44, 100 nm	1, 2, 10 µg/mL, *B*	Duke scientific corporation (Palo Alto, CA, USA)
	Barshtein [60]	polystyrene	~50, 110, 245 nm	0.05–0.5 mg/mL, *B*	Polysciences Inc. (Warrington, PA, USA)
	Walczak [69]	polystyrene	50 nm	125 mg/kg; 25 mg/mL, *A*	Magsphere (Pasadena, CA, USA), Polysciences Inc. (Warrington, PA, USA)
2012	Campbell [7]	polystyrene	20, 100, 200 nm	4 mg/mL, *B*	Invitrogen Ltd. Thermo Fisher Scientific (Waltham, MA, USA)
2004	Alvarez-Román [6]	polystyrene	20, 200 nm	0.1 mL Suspension/ 0.8 cm^2^, *C*	Thermo Fisher Scientific (Waltham, MA, USA)
	Xu [13]	Min-U-Sil	0.5–3 μm	10 mg/kg, *A*	in-house
		polyvinyl chloride/E3	0.2–2 μm	10, 50 mg/kg, *A*	APME (Munich, Germany)
		polyvinyl chloride/W3	0.2–2 μm	10, 50 mg/kg, *A*	APME (Munich, Germany)
	Rejman [12]	latex	50–1000 nm	no further details, *B*	Polysciences Inc. (Warrington, PA, USA)
2001	Brown [22]	polystyrene	64, 202, 535 nm	1 mg/mL, *A, B*	Polysciences Inc. (Warrington, PA, USA)
1998	Ogawara [64]	polystyrene	50, 500 nm	12.5 mg/kg, *A*	Polysciences Inc. (Warrington, PA, USA)
1975	Volkheimer [31]	polyvinyl chloride	5–110 μm	2.4 × 10^9^, *A*	-

*A* = in vivo, *B* = in vitro, *C* = ex vivo, TBCTRC = Tianjin Baseline ChromTech Research Center, APME = Association of Plastics Manufacturers Europe.

## Data Availability

Not applicable.

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
