# Peer review of "Cellular and Systemic Effects of Micro- and Nanoplastics in Mammals—What We Know So Far"

_materials, 2023, doi:10.3390/ma16083123_

Round 1

Reviewer 1 Report

materials-2267318-comments

The manuscript “Cellular and systemic effects of micro- and nanoplastics in mammalian – what we know so far” describes some of the published literature that have addressed the hazards posed to mammalian systems from micro- and nanoplastics. The article is well written and contains some relevant information, but I feel it falls short in the presence of numerous reviews conducted on micro- and nanoplastics. This is a very busy field with many reviews coming out recently, and there is not enough originality for this to add to the current published work. Furthermore, it is possible that there may be a lot of published work has been missed.

Minor comments:

·       Maybe the authors could use abbreviations for microplastics and nanoplastics?

·       Line 56: the authors only use microplastics in the sentence “In recent years, studies on the effects of microplastics…………..”, should be clear they also consider nanoplastics?

·       The search methods need to be better described, there must have been other terminology used to narrow-down the thousands of hits that would have received, could these be provided?

·       Line 72: again the authors use only microplastic, please be clear if you are considering both micro- and nanoplastics.

·       Line 82-83: the authors state the size limit for skin penetration is 100nm, please check this as it could be as low as 5nm, I believe there are papers that state this.

The Major comments:

·       Line 72: the authors say there are no studies that have assessed dermal transfer of microplastics, this is not true (but please bear in mind that I’m assuming they include here nanoplastics and in mammalian skin?). This transgression comes from a flaw in their search strategy, there is a lot of content when looking for e.g. "polystyrene nanoparticles" AND "dermal penetration".

·       There has been a lot of work previously performed that have measured mammalian cell and system responses that have not been included by the authors. Maybe this is because in early studies the focus was not on the material being a nanoplastic per se, but instead a nanoparticle made of PS for example. I suggest the authors try to identify some of these studies, this is the reason for the missed studies for dermal penetration (mentioned previously) but relates to other missed studies. a few examples are given here: (https://doi.org/10.1006/taap.2001.9240; https://doi.org/10.1016/j.taap.2009.09.015; https://doi.org/10.1016/j.tiv.2015.11.006). Please bear in mind that these are only a fraction of what has potentially been missed. Maybe the authors can adjust/expand the search strategy to recover potentially lost studies.

·       Although the manuscript provides a well written, interesting review of the hazard that micro- and nanoplastics may pose, it would be useful to provide a bit more interpretation. E.g. to provide an indication of what properties pose the greatest hazard. This aligns with my earlier comment that the review doesn’t provide much novelty amongst an area that is receiving a lot of reviews at present.

·       I do not think the review goes enough into the other factors posed as hazards associated with micro- and nanoplastics. The authors do have a short section on the adsorption of hazardous substances and the presence of chemical contaminants, but I feel this warrants more focus as it is a crucial aspect of micro- and nanoplastic risks.

Reviewer 2 Report

The manuscript gives a comprehensive overview of the uptake of micro- and nanoplastics and crossing of micro- and nanoplastics well-controlled barriers of the body on the following topics: micro- and nanoplastics uptake via the skin, micro- and nanoplastics uptake by inhalation, micro- and nanoplastics uptake by ingestion, impact of micro- and nanoplastics on immune cells, micro- and nanoplastics can cross well-controlled barriers of the body and general limitations of studies with micro- and nanoplastics. 

Micro- and nanoplastics are increasing and widespread use in various consumer and industrial products, thus their potential risk to the environment including living organisms both in soil and water has attracted growing concerns. The micro- and nanoplastics risk to mammalian is a recently started study and the information is scattered. To my knowledge, no comprehensive summarization has been done in this field so far. Therefore, this is a timely and excellent review that contains a considerable number of references and presents a comprehensively deep summary of micro- and nanoplastics associated with mammalian. 

The information summarized in this paper will guide and benefit both of researchers in the field of environmental toxicology, especially micro- and nanoplastics-induced toxicity to human. Also, this paper is well organized and well written. The key unsolved questions raised in this paper are very impressive and useful. I’m very glad to read this review in advance and strongly recommend it for publication after minor revision. 

My major comment is regarding the section 3. In this section, the authors review the advances of the uptake of micro- and nanoplastics via skin, inhalation and ingestion, the behavior of micro- and nanoplastics crossing the well-controlled barriers of the body, and effects of micro- and nanoplastics on immune cells. Besides effects of micro- and nanoplastics on immune cells, other contents are about the uptake and transport of micro- and nanoplastics. I suggest the authors to present “effects of micro- and nanoplastics on immune cells” at the end.

Round 2

Reviewer 1 Report

I'm happy with revision.